# Hi-Mamba: Hierarchical Mamba for Efficient Image Super-Resolution

## Abstract

State Space Models (SSM), such as Mamba, have shown strong representation ability in modeling long-range dependency with linear complexity, achieving successful applications from high-level to low-level vision tasks. However, SSM's sequential nature necessitates multiple scans in different directions to compensate for the loss of spatial dependency when unfolding the image into a 1D sequence. This multi-direction scanning strategy significantly increases the computation overhead and is unbearable for high-resolution image processing. To address this problem, we propose a novel Hierarchical Mamba network, namely, Hi-Mamba, for image super-resolution (SR). Hi-Mamba consists of two key designs: (1) The Hierarchical Mamba Block (HMB) assembled by a Local SSM (L-SSM) and a Region SSM (R-SSM) both with the single-direction scanning, aggregates multi-scale representations to enhance the context modeling ability. (2) The Direction Alternation Hierarchical Mamba Group (DA-HMG) allocates the isomeric single-direction scanning into cascading HMBs to enrich the spatial relationship modeling. Extensive experiments demonstrate the superiority of Hi-Mamba across five benchmark datasets for efficient SR. For example, Hi-Mamba achieves a significant PSNR improvement of 0.29 dB on Manga109 for $\times 3$ SR, compared to the strong lightweight MambaIR.

## 1 Introduction

Single Image Super-Resolution Yang et al. (2019); He et al. (2019); Zhang et al. (2018); Chen et al. (2022); Zhang et al. (2021) (SISR) aims to restore an authentic high-resolution (HR) image from a single degraded low-resolution (LR) one, which benefits plentiful downstream applications such as magnetic resonance imaging (MRI), mobile device photography, and video surveillance. Various studies have proposed Convolutional Neural Networks (CNNs) Ahn et al. (2018); Li et al. (2021b); Zhang et al. (2019b) to learn a mapping from LR inputs to HR outputs. Despite their efficacy and remarkable advances in the past, CNN-based SR models are reaching their upper-performance limits even with continuously increasing model sizes, due to CNNs' limited capability on long-range dependency modeling.

Transformer-based SR methods Liang et al. (2021); Chen et al. (2023b;a); Ray et al. (2024); Zhang et al. (2024) introduce self-attention mechanisms with extraordinary long-range modeling capabilities to remarkably improve SR performance, while at the cost of quadratic computational complexity. Numerous subsequent works have been proposed to make the vanilla Transformers more efficient and powerful via shifted window attention Liang et al. (2021); Zhang et al. (2022b), transposed attentions Zamir et al. (2022); Li et al. (2023b) and anchored stripe self-attention Li et al. (2023c), *etc.* However, these studies are difficult to relieve the quadratic complexity of attention mechanisms at inference in practice.

Recently, Mamba Gu & Dao (2024) architecture constructed on Structured State Space Models (S4) has emerged as a promising technique due to its high potential in long-sequence modeling with linear complexity. As S4 was originally proposed in the field of natural language processing (NLP) Gu et al. (2021a); Gu & Dao (2024), several succeeding works have introduced S4 into vision recognition tasks Liu et al. (2024); Zhu et al. (2024) and image processing tasks Shi et al. (2024), demonstrating impressive results. For example, Vision Mamba Zhu et al. (2024) was proposed for image recognition tasks, manifesting that Vision Mamba can overcome the computation & memory constraints on

Table 1: Comparison of different scanning modes in MambaIR for $\times 2$ SR. MambaIR-$n$ indicates using the number of $n$ scanning.

| Method | GPU(ms) | Params | FLOPs | Set5 | Set14 | B100 | Urban100 | Manga109 |
|---|---|---|---|---|---|---|---|---|
| MambaIR-1 Guo et al. (2024) | 519 | 987K | 291G | 38.13 | 33.86 | 32.31 | 32.82 | 39.19 |
| MambaIR-2 Guo et al. (2024) | 653 | 1.11M | 383G | 38.15 | 33.94 | 32.31 | 32.86 | 39.26 |
| MambaIR-4 Guo et al. (2024) | 982 | 1.36M | 568G | 38.16 | 34.00 | 32.34 | 32.92 | 39.31 |
| Hi-Mamba-S (Ours) | 379 | 1.34M | 274G | 38.24 | 34.08 | 32.38 | 33.13 | 39.35 |

image perceptions. For low-level vision tasks, MambaIR Guo et al. (2024) introduces the vision state-space module (VSSM) from Vmamba Liu et al. (2024) for image super-resolution and achieves performance comparable to Transformer-based SR baselines.

Previous vision Mamba architectures typically employ a multi-direction scanning strategy to compensate for the loss of spatial dependencies when unfolding the image into a 1D sequence. Unfortunately, the repetitive multiple-sequence scanning overshadows the essential linear computational complexity of SSMs primarily designed with single-sequence scanning to model 1D sequential relationships. It significantly increases the computation overhead and is unacceptable for high-resolution image processing tasks. As shown in Tab. 1, the four-sequence scanning approach effectively improves the performance by 0.10 dB and 0.12 dB on Urban100 and Manga109, respectively. However, this enhancement comes at a significant computational cost, increasing FLOPs by 95.2% and parameters by 37.8% compared to the single-sequence scanning approach in MambaIR.

To address this problem, we propose a novel Hierarchical Mamba architecture, termed Hi-Mamba, for image super-resolution (SR). We first propose the hierarchical Mamba block (HMB) which is constructed by a local SSM and a region SSM with single-direction scanning to conduct multi-scale data-dependent visual context modeling. Furthermore, we propose the direction alternation hierarchical Mamba group (DA-HMG) that allocates the isomeric single-direction scanning into cascaded HMBs to enrich the spatial relationship modeling. Our DA-HMG improves the reconstruction performance with no extra FLOPs or parameter increases. In addition, we propose that the gate feed-forward network (G-FFN) introduce additional non-linear information through a simple gate mechanism in the feed-forward network. We verify the effectiveness of Hi-Mamba on several classical SR benchmarks with three released versions, which makes fair comparisons with various SR models with different capacities.

We summarize our main contributions as follows:

- We propose Hi-Mamba for efficient SR, incorporating hierarchical Mamba block (HMB), specifically the Local-SSM and the Region-SSM for multi-scale data-dependent visual context modeling.

- The direction alternation hierarchical Mamba group (DA-HMG) is simple yet effective in enriching the spatial relationship modeling, which allocates the isomeric single-direction scanning into cascaded HMBs to improve performance without incurring extra computation and memory costs.

- Extensive experiments demonstrate the superiority of the proposed Hi-Mamba. For example, our Hi-Mamba achieves significant PSNR gains of 0.37dB on Urban100 for $\times 3$ SR compared to SRFormer Zhou et al. (2023).

## 2 RELATED WORK

### 2.1 EFFICIENT CNNS AND TRANSFORMERS FOR SUPER-RESOLUTION

Since SRCNNDong et al. (2015) first introduced convolutional neural networks (CNNs) for SR, various works Dong et al. (2016); Lim et al. (2017); Ledig et al. (2017); Zhang et al. (2018) have explored CNN-based SR architectures to improve SR performance. To improve model efficiency, CARN Ahn et al. (2018) proposes a cascading mechanism at both the local and global levels. IMDN Hui et al. (2019) adopts feature splitting and concatenation operations to progressively aggregate features, further reducing parameters. SAFMN Sun et al. (2023) utilizes a feature pyramid to generate spatially-adaptive feature attention maps. However, these CNN-based SR methods are limited by the size of the convolutional kernels and cannot effectively model long-term dependencies between pixels.

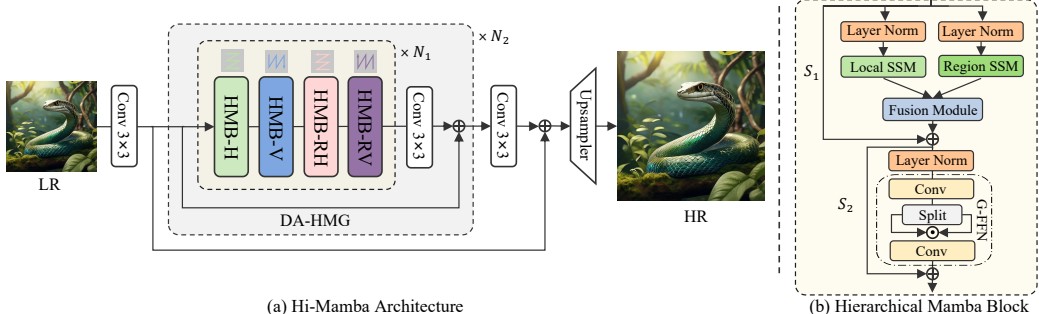

<p style="text-align:center">(a) Hi-Mamba Architecture            (b) Hierarchical Mamba Block</p>

Figure 1: Illustration of the proposed Hi-Mamba. (a) The overview of Hi-Mamba architecture with $N_2$ Hierarchical Mamba Groups (DA-HMG), where each DA-HMG contains the number of $N_1$ Hierarchical Mamba blocks (HMB), which consist of four isomeric single-direction scanning SSM denoted by HMB-H/V/RH/RV. (b) Hierarchical Mamba Block (HMB) consists of a Local-SSM, a Region-SSM, and a Gate Feed-Forward Network (G-FFN).

To capture long-range pixel dependencies, Transformer-based methods Liang et al. (2021); Chen et al. (2023b;a); Ray et al. (2024); Zhang et al. (2024) have introduced self-attention mechanisms into SR tasks, achieving significant performance improvements. To facilitate practical deployment, various efficient attention mechanisms Li et al. (2023c); Zhou et al. (2023) have been proposed to reduce computational and memory costs. ESRT Zhisheng et al. (2021) computes attention maps in a group manner to reduce memory usage. N-Gram Choi et al. (2023) proposed an asymmetric U-Net architecture that downsamples features to reduce computational cost. DLGSANet Li et al. (2023b) utilizes channel-wise self-attention, which has lower computational costs compared to spatial self-attention. SRFormer Zhou et al. (2023) minimizes the size of the attention map by compressing the channel dimensions of the key and value in self-attention. However, these methods do not directly address the quadratically growing complexity of attention mechanisms with the increase in token sequence length. Moreover, they typically compute self-attention based on windows, which confines the receptive field for high-quality image reconstruction.

## 2.2 MAMBA AND APPLICATIONS FOR SUPER-RESOLUTION

State space models (SSM) Gu et al. (2021a;b); Smith et al. (2022), originating from classical control theory Kalman (1960), are rising as novel backbones in Deep Learning. Successful applications of SSM include Mamba Gu & Dao (2024), Vim Zhu et al. (2024), and VMamba Liu et al. (2024), which are all tailored toward high-level image understanding tasks. Overall, the implementations of SSM in low-level vision tasks remain few. MambaIR Guo et al. (2024) first introduced the Mamba architecture to image super-resolution tasks, achieving impressive image restoration results. MMA Cheng et al. (2024) introduced Vision Mamba (ViM) Zhu et al. (2024) and combined it with convolutional structures to activate a wider pixel area, thereby enhancing SR performance. DVMSR Lei et al. (2024) was the first to attempt distilling the Mamba architecture to achieve an ultra-lightweight SR Mamba model. FMSR Xiao et al. (2024) introduced Mamba for remote sensing image super-resolution, which uses frequency information to assist the Mamba architecture, achieving performance surpassing Transformer methods. However, these methods all use multi-sequence scanning strategies to model the image spatial relationships, which significantly increases computational costs compared to the single-sequence scanning of vanilla Mamba. Different from these methods, our Hi-Mamba uses only single-sequence scanning and proposes HMB to compensate for SSM's inadequacy in modeling 2D-pixel relationships. Additionally, the DA-HMG is proposed to enrich spatial relationship modeling by alternatively changing the single-sequence scanning direction in HMB without additional computational costs.

## 3 HIERARCHICAL MAMBA NETWORKS

### 3.1 PRELIMINARIES

SSM can be viewed as a Linear Time-Invariant (LTI) system, which maps the input one-dimensional function or sequence $x(t) \in \mathbb{R}$ to the output response $y(t) \in \mathbb{R}$ through a hidden state $h(t) \in \mathbb{R}^N$.

They are typically represented as linear ordinary differential equations:

$$h'(t) = Ah(t) + Bx(t), \quad y(t) = Ch(t) + Dx(t), \tag{1}$$

where $A \in \mathbb{R}^{N \times N}$, $B \in \mathbb{R}^{N \times 1}$, $C \in \mathbb{R}^{1 \times N}$, and $D \in \mathbb{R}$ are weight parameters, and $N$ represents the state size.

The discretization process is commonly used to process Eq. 1, which can be applied in deep learning scenarios. In particular, the timescale parameter $\Delta$ is used to convert the continuous parameters $A$ and $B$ into discrete ones $\overline{A}$ and $\overline{B}$. The widely used discretization method adheres to the Zero-Order Hold (ZOH) rule, which is formulated as:

$$\overline{A} = \exp(\Delta A), \quad \overline{B} = (\Delta A)^{-1}(\exp(A) - I) \cdot \Delta B. \tag{2}$$

Therefore, after discretization, Eq. 1 can be rewritten as:

$$h_k = \overline{A}h_{k-1} + \overline{B}x_k, \quad y_k = Ch_k + Dx_k. \tag{3}$$

To further accelerate computation, Gu et al. Gu et al. (2021a) expanded the SSM computation into a convolution with a structured convolutional kernel $\overline{K} \in \mathbb{R}^L$:

$$\overline{K} \triangleq \left( C\overline{B}, C\overline{AB}, \cdots, C\overline{A}^{L-1}\overline{B} \right), \quad y = x * \overline{K}, \tag{4}$$

where $L$ is the length of the input sequence and $*$ denotes the convolution operation. A recent state space model, Mamba Gu & Dao (2024), introduces Selective State Space Models (S6) by relaxing the time-invariance constraints on $B$, $C$, and $\Delta$ depending on the input $x$, which selectively propagates information for 1D language sequence modeling.

To expand Mamba from 1D language sequences to 2D visual inputs, various works Liu et al. (2024); Liang et al. (2024); Deng & Gu (2024); Guo et al. (2024) employ 2D selective scan (SS2D) mechanism to capture spatial correlations with 2D feature sequences. For example, VMamba Liu et al. (2024) employs SS2D by scanning four directed input sequences and generating the 2D feature map by independently combining four feature sequences via an S6 block. Similarly, MambaIR Guo et al. (2024) introduces the Vision State-Space Module (VSSM) into image restoration for information interaction at the whole-image level. However, these methods employ repetitive multi-direction scanning to adapt to 2D image inputs, significantly increasing computational costs.

## 3.2 ARCHITECTURE OVERVIEW

As shown in Fig. 1 (a), the proposed Hierarchical Mamba (Hi-Mamba) architecture comprises three parts: shallow feature extraction, deep feature extraction, and image reconstruction. Given a low-resolution (LR) input image $I_{\text{LR}} \in \mathbb{R}^{C_{\text{in}} \times H \times W}$, where $C_{\text{in}}$, $H$, and $W$ are the input channels, height, and width, respectively. We first use a simple convolution for shallow feature extraction $H_{SF}$ to generate local features $F_l \in \mathbb{R}^{C \times H \times W}$:

$$F_l = H_{SF}(I_{LR}), \tag{5}$$

where $C$ is the embedding channel dimension. Subsequently, the local features $F_l$ are processed in the deep feature extraction module $H_{DF}$ to obtain deep features $F_d \in \mathbb{R}^{C \times H \times W}$:

$$F_d = H_{DF}(F_l), \tag{6}$$

where the deep feature extraction module $H_{DF}$ consists of multiple direction alternation hierarchical Mamba groups (DA-HMG) with a total number of $N_2$. To ensure training stability, a residual strategy is adopted within each group. Each DA-HMG contains the number of $N_1$ Hierarchical Mamba blocks (HMB), which consist of four isomeric single-direction scanning SSM denoted by HMB-H/V/RH/RV. At the end of each DA-HMG, convolutional layers are introduced to refine the features.

Finally, we use $F_l$ and $F_d$ as the inputs and reconstruct the high-resolution (HR) output image $H_R$ through the reconstruction module, which can be formulated as:

$$I_r = H_R(F_l + F_d), \tag{7}$$

where $H_R$ involves a single 3×3 convolution followed by a pixel shuffle operation. We optimize the parameters $\theta$ of Hi-Mamba by the pixel-wise L1 loss between the reconstruction output $I_r$ and the ground truth (GT) $I_{gt}$. In the following, we will introduce the key blocks and modules in Hi-Mamba.

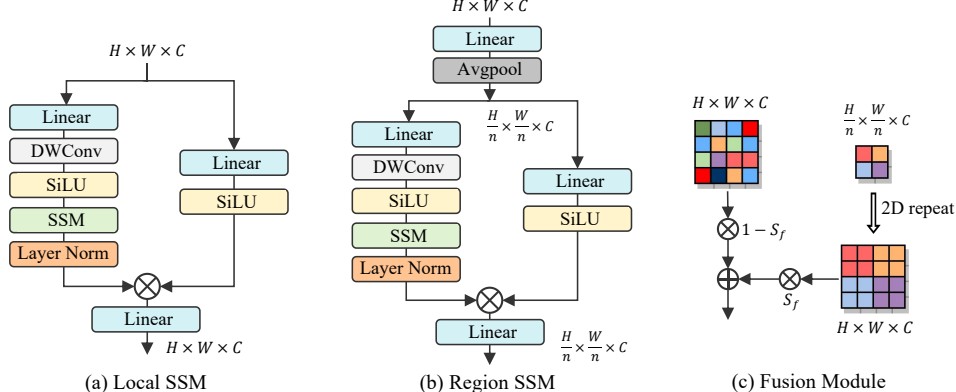

Figure 2: Illustration of the key components in HMB.

### 3.3 HIERARCHICAL MAMBA BLOCK

The original visual Mamba blocks Liu et al. (2024); Zhu et al. (2024) typically employ multi-direction scanning, which significantly increases the computation overhead. To address this problem, we design a novel hierarchical mamba block (HMB) with only single-direction scanning and alternatively change the scanning direction to enrich the spatial relationship modeling to construct DA-HMG.

As illustrated in Fig. 1 (b), HMB primarily consists of two branches: Local SSM (L-SSM) and Region SSM (R-SSM). Given the local input feature $I_l^i \in \mathbb{R}^{C \times H \times W}$ and the region input feature $I_r^i \in \mathbb{R}^{C \times \frac{H}{n} \times \frac{W}{n}}$ at the $i$-th layer, we first employ Layer Normalization (LN) and go through two branches to capture long-range dependencies. Additionally, we incorporate learnable scaling factors $S_1 \in \mathbb{R}^C$ to regulate the information within skip connections:

$$F_l^i = \text{L-SSM}(\text{LN}(I_l^i)), F_r^i = \text{R-SSM}(\text{LN}(I_r^i)),$$
$$F^i = (F_l^i \otimes F_r^i) + (S_1 \cdot I_l^i). \tag{8}$$

where $F_l^i$, and $F_r^i$ are the outputs of these two branches, respectively. $\otimes$ denotes the fusion module. The L-/R-SSM and fusion modules will be described in Sec. 3.3.1 and Sec. 3.3.2, respectively.

Subsequently, the intermediate features $F^i$ will subsequently undergo the proposed gate feed-forward network (G-FFN) followed by another learnable scale factor $S_2$ in the residual connection to obtain the input features at the $i + 1$-th layer and $F_r^i$ is directly used as the regional input for the next layer:

$$F_l^{i+1} = \text{G-FFN}\left(\text{LN}(F^i)\right) + S_2 \cdot F^i, \quad F_r^{i+1} = F_r^i. \tag{9}$$

In G-FFN, we enhance the modeling capacity for spatial information by introducing a gate mechanism into the FFN. This also reduces redundant information in the channels. G-FFN first extracts features through convolution and splits the feature map along the channel dimension into two parts for element-wise multiplication. Specifically, G-FFN is computed as:

$$\hat{F} = w^1 * \text{LN}(F^i), [\hat{F}_1, \hat{F}_2] = \text{Split}(\hat{F}),$$
$$\text{G-FFN}(F^i) = w^2 * (\hat{F}_1 \odot \hat{F}_2), \tag{10}$$

where $w^1$ and $w^2$ are the convolution weights. $\odot$ is an element-wise multiplication operation. Note that, we only use a single-direction scanning in one HMB, *i.e.*, one selection from the horizontal, vertical, reverse horizontal, and reverse vertical directions.

#### 3.3.1 LOCAL / REGION SSM

Following the VSSM of MambaIR Guo et al. (2024), L-SSM and R-SSM use a similar computational sequence. Instead of VSSM with multiple-sequence scanning, L-SSM and R-SSM employ single-sequence scanning to reduce the computation costs. The architecture of L-SSM and R-SSM are illustrated in Fig. 2 (a) and (b), respectively. L-SSM and R-SSM take the local feature $I_l \in \mathbb{R}^{C \times H \times W}$ and the region feature $I_r \in \mathbb{R}^{C \times \frac{H}{n} \times \frac{W}{n}}$ as the inputs, respectively. Here, $I_r$ is generated by a simple

projection operation with a region size of $n$ to the local feature $I_l$. For simplicity, we denote the input uniformly as $X$, due to the same computation process to L-SSM and R-SSM.

In the first branch of L-SSM, feature channels are expanded to $\lambda C$ via a linear layer, where $\lambda$ is a predefined channel expansion factor, followed by depthwise convolution, SiLU Shazeer (2020) activation function, SSM and LayerNorm. In the second branch, feature channels are also expanded to $\lambda C$ with a linear layer and SiLU activation function. Finally, the features from both branches are merged and projected back to $C$ to generate an output $X_{\text{out}}$ with the same shape as the input. The above computation process can be formulated as:

$$
\begin{aligned}
X_{b1} &= \text{LN}(\text{SSM}(\text{SiLU}(\text{DWConv}(\text{Linear}(X))))), \\
X_{b2} &= \text{SiLU}(\text{Linear}(X)), \\
X_{out} &= \text{Linear}(X_{b1} \odot X_{b2}),
\end{aligned}
\tag{11}
$$

where DWConv($\cdot$), SSM($\cdot$) and $\odot$ represent depthwise convolution, SS2D Liu et al. (2024) with single-direction scanning and element-wise multiplication, respectively.

### 3.3.2 FUSION MODULE

To reinforce spatial dependencies in the 2D domains, we use the fusion module to leverage region information from adjacent pixels in the R-SSM to guide the single-sequence local feature modeling. As illustrated in Fig. 2 (c), we first repeat the region features along the spatial dimension to match the size of the local features, ensuring that each region token is mapped to the corresponding local token. This operation implicitly incorporates spatial positional information. To dynamic control the fusion results, we introduce learnable fusion scaling factors $S_f \in \mathbb{R}^C$ to fuse the outputs of L-SSM and R-SSM in Eq. 11, which is formulated as:

$$
F_{out} = S_f \cdot X_{out}^l + (1 - S_f) \cdot f_{re}(X_{out}^r).
\tag{12}
$$

where $X_{out}^l$ and $X_{out}^r$ denote the outputs of the L-SSM and R-SSM, respectively. $f_{re}$ represents the repeat operation along the 2D spatial dimension.

### 3.4 DIRECTION ALTERNATION HIERARCHICAL MAMBA GROUP

As depicted in Fig. 1, DA-HMG is easy to implement by alternatively allocating the isomeric single-direction scanning to different HMBs. By default, we apply Horizontal HMB (HMB-H), Vertical HMB (HMB-V), Reverse Horizontal HMB (HMB-RH), and Reverse Vertical HMB (HMB-RV) orders to enrich the spatial relationship modeling further. DA-HMG does not incur extra parameters and computational costs, compared to the HMB with the same direction, denoted by base-single.

Compared to the stacked multi-sequence scanning in the 2D-SSM module of MambaIR, DA-HMG significantly reduces the computational and parameter overhead while achieving superior performance. The more detailed difference between base-single, 2D-SSM and DA-HMG on the sequence scanning strategy is presented in Fig. 7 of Appendix.

## 4 EXPERIMENTS

### 4.1 EXPERIMENTAL SETTINGS

**Datasets.** Following Liang et al. (2021); Guo et al. (2024); Li et al. (2023a); Chen et al. (2023b), we train our model on two widely-used datasets, DIV2K Agustsson & Timofte (2017) and Flicker2K Lim et al. (2017), and only use DIV2K dataset to train the lightweight version of our model. We evaluate our method on five standard SR benchmarks: Set5 Bevilacqua et al. (2012), Set14 Zeyde et al. (2012), BSD100 Martin et al. (2001), Urban100 Huang et al. (2015), and Manga109 Matsui et al. (2017) across three scaling factors, $\times 2$, $\times 3$, and $\times 4$. For the evaluation metric, we calculate PSNR and SSIM Wang et al. (2004) on the Y channel in the YCbCr space and also report the average inference time (20 runs) on one NVIDIA V100, parameters and FLOPs.

**Implementation details.** Following the general setting Liang et al. (2021); Zhang et al. (2022a); Chen et al. (2022), each training sample is augmented through flipping and rotations of $90°$, $180°$ and $270°$. During training, we randomly crop images into $64 \times 64$ patches, with a total iteration number of

Table 2: Quantative comparison of lightweight SR models on five benchmarks. The best and second-best results for Transformers and Mamba are marked in red and blue colors.

| Scale | Model | Params (M) | FLOPs (G) | Set5 PSNR | Set5 SSIM | Set14 PSNR | Set14 SSIM | BSD100 PSNR | BSD100 SSIM | Urban100 PSNR | Urban100 SSIM | Manga109 PSNR | Manga109 SSIM |
|---|---|---|---|---|---|---|---|---|---|---|---|---|---|
| x2 | CARN Ahn et al. (2018) | 1.45 | 223 | 37.76 | 0.9590 | 33.52 | 0.9166 | 32.09 | 0.8978 | 31.92 | 0.9256 | 38.36 | 0.9765 |
| | EDSR-baseline Lim et al. (2017) | 1.37 | 316 | 37.99 | 0.9604 | 33.57 | 0.9175 | 32.16 | 0.8994 | 31.98 | 0.9272 | 38.54 | 0.9769 |
| | IMDN Hui et al. (2019) | 0.69 | 159 | 38.00 | 0.9605 | 33.63 | 0.9177 | 32.19 | 0.8996 | 32.17 | 0.9283 | 38.88 | 0.9774 |
| | LAPAR-A Li et al. (2020) | 0.55 | 171 | 38.01 | 0.9605 | 33.62 | 0.9183 | 32.19 | 0.8999 | 32.10 | 0.9283 | 38.67 | 0.9772 |
| | LatticeNet Luo et al. (2020) | 0.76 | 170 | 38.15 | 0.9610 | 33.25 | 0.9193 | 32.25 | 0.9005 | 32.43 | 0.9302 | - | - |
| | ESRT Zhisheng et al. (2021) | 0.67 | - | 38.03 | 0.9600 | 33.75 | 0.9184 | 32.25 | 0.9001 | 32.58 | 0.9318 | 39.12 | 0.9774 |
| | SwinIR-Light Liang et al. (2021) | 0.90 | 235 | 38.14 | 0.9611 | 33.86 | 0.9206 | 32.31 | 0.9012 | 32.76 | 0.9340 | 39.12 | 0.9783 |
| | N-Gram Choi et al. (2023) | 1.01 | 140 | 38.05 | 0.9610 | 33.79 | 0.9199 | 32.27 | 0.9008 | 32.53 | 0.9324 | 38.97 | 0.9777 |
| | SRFormer-Light Zhou et al. (2023) | 0.83 | 236 | 38.23 | 0.9613 | 33.94 | 0.9209 | 32.36 | 0.9019 | 32.91 | 0.9353 | 39.28 | 0.9785 |
| | MambaIR Guo et al. (2024) | 1.36 | 568 | 38.16 | 0.9610 | 34.00 | 0.9212 | 32.34 | 0.9017 | 32.92 | 0.9356 | 39.31 | 0.9779 |
| | Hi-Mamba-T | 0.87 | 178 | 38.24 | 0.9613 | 34.06 | 0.9215 | 32.35 | 0.9019 | 33.04 | 0.9358 | 39.28 | 0.9785 |
| | Hi-Mamba-S | 1.34 | 274 | 38.24 | 0.9614 | 34.08 | 0.9217 | 32.38 | 0.9021 | 33.13 | 0.9368 | 39.35 | 0.9788 |
| x3 | CARN Ahn et al. (2018) | 1.59 | 119 | 34.29 | 0.9255 | 30.29 | 0.8407 | 29.06 | 0.8034 | 28.06 | 0.8493 | 33.50 | 0.9440 |
| | EDSR-baseline Lim et al. (2017) | 1.56 | 160 | 34.37 | 0.9270 | 30.28 | 0.8417 | 29.09 | 0.8052 | 28.15 | 0.8527 | 33.45 | 0.9439 |
| | IMDN Hui et al. (2019) | 0.70 | 72 | 34.36 | 0.9270 | 30.32 | 0.8417 | 29.09 | 0.8046 | 28.17 | 0.8519 | 33.61 | 0.9445 |
| | LAPAR-A Li et al. (2020) | 0.54 | 114 | 34.36 | 0.9267 | 30.34 | 0.8421 | 29.11 | 0.8054 | 28.15 | 0.8523 | 33.51 | 0.9441 |
| | LatticeNet Luo et al. (2020) | 0.77 | 76 | 34.53 | 0.9281 | 30.39 | 0.8424 | 29.15 | 0.8059 | 28.33 | 0.8538 | - | - |
| | ESRT Zhisheng et al. (2021) | 0.77 | - | 34.42 | 0.9268 | 30.43 | 0.8433 | 29.15 | 0.8063 | 28.46 | 0.8574 | 33.95 | 0.9455 |
| | SwinIR-Light Liang et al. (2021) | 0.89 | 87 | 34.62 | 0.9289 | 30.54 | 0.8463 | 29.20 | 0.8082 | 28.66 | 0.8624 | 33.98 | 0.9478 |
| | N-Gram Choi et al. (2023) | 1.01 | 67 | 34.52 | 0.9282 | 30.53 | 0.8456 | 29.19 | 0.8078 | 28.52 | 0.8603 | 33.89 | 0.9470 |
| | SRFormer-Light Zhou et al. (2023) | 0.86 | 105 | 34.67 | 0.9296 | 30.57 | 0.8469 | 29.26 | 0.8099 | 28.81 | 0.8655 | 34.19 | 0.9489 |
| | MambaIR Guo et al. (2024) | 1.37 | 253 | 34.72 | 0.9296 | 30.63 | 0.8475 | 29.29 | 0.8099 | 29.00 | 0.8689 | 34.39 | 0.9495 |
| | Hi-Mamba-T | 0.88 | 80 | 34.76 | 0.9298 | 30.61 | 0.8472 | 29.27 | 0.8091 | 29.05 | 0.8693 | 34.42 | 0.9499 |
| | Hi-Mamba-S | 1.35 | 123 | 34.77 | 0.9303 | 30.68 | 0.8493 | 29.33 | 0.8111 | 29.18 | 0.8716 | 34.68 | 0.9509 |
| x4 | CARN Ahn et al. (2018) | 1.59 | 91 | 32.13 | 0.8937 | 28.6 | 0.7806 | 27.58 | 0.7349 | 26.07 | 0.7837 | 30.47 | 0.9084 |
| | EDSR-baseline Lim et al. (2017) | 1.52 | 114 | 32.09 | 0.8938 | 28.58 | 0.7813 | 27.57 | 0.7357 | 26.04 | 0.7849 | 30.35 | 0.9067 |
| | IMDN Hui et al. (2019) | 0.72 | 41 | 32.21 | 0.8948 | 28.58 | 0.7811 | 27.56 | 0.7353 | 26.04 | 0.7838 | 30.45 | 0.9075 |
| | LAPAR-A Li et al. (2020) | 0.66 | 94 | 32.15 | 0.8944 | 28.61 | 0.7818 | 27.61 | 0.7366 | 26.14 | 0.7871 | 30.42 | 0.9074 |
| | LatticeNet Luo et al. (2020) | 0.78 | 44 | 32.30 | 0.8962 | 28.68 | 0.7830 | 27.62 | 0.7367 | 26.25 | 0.7873 | - | - |
| | ESRT Zhisheng et al. (2021) | 0.75 | 64 | 32.19 | 0.8947 | 28.69 | 0.7833 | 27.69 | 0.7379 | 26.39 | 0.7962 | 30.75 | 0.9100 |
| | SwinIR-Light Liang et al. (2021) | 0.90 | 50 | 32.44 | 0.8976 | 28.77 | 0.7858 | 27.69 | 0.7406 | 26.47 | 0.7980 | 30.92 | 0.9151 |
| | N-Gram Choi et al. (2023) | 1.02 | 36 | 32.33 | 0.8963 | 28.78 | 0.7859 | 27.66 | 0.7396 | 26.45 | 0.7963 | 30.80 | 0.9128 |
| | SRFormer-Light Zhou et al. (2023) | 0.87 | 63 | 32.51 | 0.8988 | 28.82 | 0.7872 | 27.73 | 0.7422 | 26.67 | 0.8032 | 31.17 | 0.9165 |
| | MambaIR Guo et al. (2024) | 1.40 | 143 | 32.51 | 0.8993 | 28.82 | 0.7876 | 27.65 | 0.7423 | 26.75 | 0.8051 | 31.26 | 0.9175 |
| | Hi-Mamba-T | 0.89 | 45 | 32.52 | 0.8995 | 28.80 | 0.7873 | 27.75 | 0.7429 | 26.81 | 0.8072 | 31.35 | 0.9186 |
| | Hi-Mamba-S | 1.36 | 69 | 32.60 | 0.8999 | 28.91 | 0.7895 | 27.78 | 0.7436 | 26.86 | 0.8086 | 31.46 | 0.9192 |

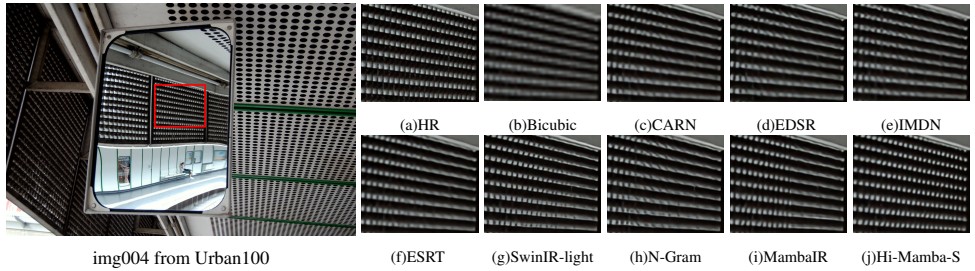

(a)HR   (b)Bicubic   (c)CARN   (d)EDSR   (e)IMDN

img004 from Urban100   (f)ESRT   (g)SwinIR-light   (h)N-Gram   (i)MambaIR   (j)Hi-Mamba-S

Figure 3: Qualitative comparison on the "img004" image of Urban100 for ×4 SR.

500K. The patch size is set to 32. We employ the Adam optimizer with training parameters $\beta_1 = 0.9$, $\beta_2 = 0.999$, and zero weight decay. The initial learning rate was 2e-4, which was halved at iterations [250K, 400K, 450K, 475K]. The experiments are implemented by PyTorch using 8 NVIDIA V100 GPUs. We provide three versions of Hi-Mamba with varying complexities, denoted as Hi-Mamba-T, Hi-Mamba-S and Hi-Mamba-L. The details of the three versions can be found in the Appendix.

## 4.2 COMPARISON WITH LIGHTWEIGHT SR MODELS.

**Quantitative evaluations.** Tab. 2 summarizes the quantitative results at three SR scale factors of ×2, ×3 and ×4. The parameter and computational costs of MambaIRGuo et al. (2024) are modified by the tool[1]Compared to CNN-based methods, Transformer-based approaches (such as IMDN Hui et al. (2019) and SRFormer-Light Zhou et al. (2023)) introduce self-attention mechanisms to model long-range dependencies, exhibiting superior performance in terms of PSNR and SSIM. Notably, transformer-based methods often utilize window-based self-attention mechanisms in the super-resolution task to reduce computational but limit the receptive field within the window. In contrast, MambaIR employs the SSM to model long-range dependencies, which outperforms the SOTA SRFormer Zhou et al. (2023) by 0.09 PSNR on Urban100 for 3× SR. However, MambaIR requires 1.59× parameter and 2.41× FLOPs compared to SRFormer. This is due to the usage of computation-heavy multi-sequence directional scanning in SSM and the redundant structural design. For a fair comparison, we compare the proposed Hi-Mamba-T and Hi-Mamba-S with state-of-the-art

---

[1]https://github.com/MzeroMiko/VMamba/blob/main/classification/models/vmamba.py#L1372

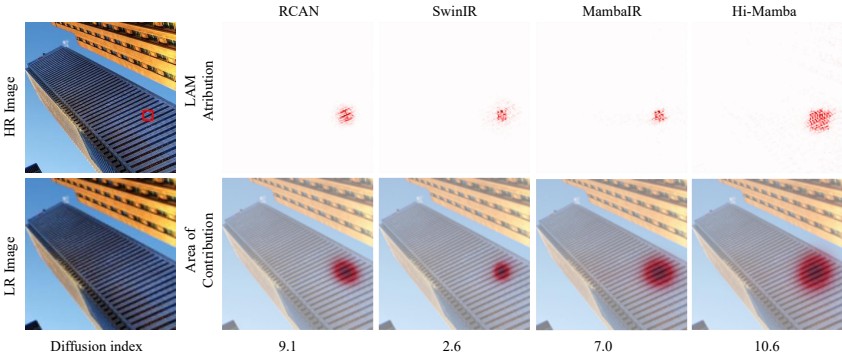

Figure 4: LAM visualization Gu & Dong (2021) on ×2 SR task. LAM indicates the correlation between the significance of each pixel in LR and the SR patch outlined with the red box. Hi-Mamba utilizes a broader range of information to obtain better performance.

Table 3: Comparison of different PSNR-oriented SR models on five benchmarks. Methods with "*" are replicated with standard setting, detailed in the Appendix. Methods with "+" denote the use of a self-ensemble strategy.

| Model | Scale | Set5 | | Set14 | | BSD100 | | Urban100 | | Manga109 | |
|---|---|---|---|---|---|---|---|---|---|---|---|
| | | PSNR | SSIM | PSNR | SSIM | PSNR | SSIM | PSNR | SSIM | PSNR | SSIM |
| EDSR Lim et al. (2017) | ×2 | 38.11 | 0.9602 | 33.92 | 0.9195 | 32.32 | 0.9013 | 32.93 | 0.9351 | 39.10 | 0.9773 |
| RCAN Zhang et al. (2018) | ×2 | 38.27 | 0.9614 | 34.12 | 0.9216 | 32.41 | 0.9027 | 33.34 | 0.9384 | 39.44 | 0.9786 |
| SAN Dai et al. (2019) | ×2 | 38.31 | 0.9620 | 34.07 | 0.9213 | 32.42 | 0.9028 | 33.10 | 0.9370 | 39.32 | 0.9792 |
| HAN Niu et al. (2020) | ×2 | 38.27 | 0.9614 | 34.16 | 0.9217 | 32.41 | 0.9027 | 33.35 | 0.9385 | 39.46 | 0.9785 |
| IGNN Zhou et al. (2020) | ×2 | 38.24 | 0.9613 | 34.07 | 0.9217 | 32.41 | 0.9025 | 33.23 | 0.9383 | 39.35 | 0.9786 |
| CSNLN Mei et al. (2020) | ×2 | 38.28 | 0.9616 | 34.12 | 0.9223 | 32.40 | 0.9024 | 33.25 | 0.9386 | 39.37 | 0.9785 |
| NLSN Mei et al. (2021) | ×2 | 38.34 | 0.9618 | 34.08 | 0.9231 | 32.43 | 0.9027 | 33.42 | 0.9394 | 39.59 | 0.9789 |
| ELAN Zhang et al. (2022b) | ×2 | 38.36 | 0.9620 | 33.20 | 0.9228 | 32.45 | 0.9030 | 33.44 | 0.9391 | 39.62 | 0.9793 |
| DLGSANet Li et al. (2023b) | ×2 | 38.34 | 0.9617 | 34.25 | 0.9231 | 32.38 | 0.9025 | 33.41 | 0.9393 | 39.57 | 0.9789 |
| IPT Chen et al. (2021) | ×2 | 38.37 | - | 34.43 | - | 32.48 | - | 33.76 | - | - | - |
| SwinIR Liang et al. (2021) | ×2 | 38.42 | 0.9623 | 34.46 | 0.9250 | 32.53 | 0.9041 | 33.81 | 0.9427 | 39.92 | 0.9797 |
| EDT Li et al. (2021a) | ×2 | 38.45 | 0.9624 | 34.57 | 0.9258 | 32.52 | 0.9041 | 33.80 | 0.9425 | 39.93 | 0.9800 |
| GRL-B* Li et al. (2023c) | ×2 | 38.48 | 0.9627 | 34.64 | 0.9265 | 32.55 | 0.9045 | 33.97 | 0.9437 | 40.06 | 0.9804 |
| SRFormer Zhou et al. (2023) | ×2 | 38.51 | 0.9627 | 34.44 | 0.9253 | 32.57 | 0.9046 | 34.09 | 0.9449 | 40.07 | 0.9802 |
| MambaIR Guo et al. (2024) | ×2 | 38.57 | 0.9627 | 34.67 | 0.9261 | 32.58 | 0.9048 | 34.15 | 0.9466 | 40.28 | 0.9806 |
| **Hi-Mamba-L** | ×2 | 38.58 | 0.9633 | 34.70 | 0.9264 | 32.60 | 0.9054 | 34.22 | 0.9475 | 40.38 | 0.9820 |
| **Hi-Mamba-L+** | ×2 | 38.60 | 0.9634 | 34.78 | 0.9269 | 32.63 | 0.9058 | 34.34 | 0.9483 | 40.49 | 0.9822 |
| EDSR Lim et al. (2017) | ×4 | 32.46 | 0.8968 | 28.80 | 0.7876 | 27.71 | 0.7420 | 26.64 | 0.8033 | 31.02 | 0.9148 |
| RCAN Zhang et al. (2018) | ×4 | 32.63 | 0.9002 | 28.87 | 0.7889 | 27.77 | 0.7436 | 26.82 | 0.8087 | 31.22 | 0.9173 |
| SAN Dai et al. (2019) | ×4 | 32.64 | 0.9003 | 28.92 | 0.7888 | 27.78 | 0.7436 | 26.79 | 0.8068 | 31.18 | 0.9169 |
| HAN Niu et al. (2020) | ×4 | 32.64 | 0.9002 | 28.90 | 0.7890 | 27.80 | 0.7442 | 26.85 | 0.8094 | 31.42 | 0.9177 |
| IGNN Zhou et al. (2020) | ×4 | 32.57 | 0.8998 | 28.85 | 0.7891 | 27.77 | 0.7434 | 26.84 | 0.8090 | 31.28 | 0.9182 |
| CSNLN Mei et al. (2020) | ×4 | 32.68 | 0.9004 | 28.95 | 0.7888 | 27.80 | 0.7439 | 27.22 | 0.8168 | 31.43 | 0.9201 |
| NLSN Mei et al. (2021) | ×4 | 32.59 | 0.9000 | 28.87 | 0.7891 | 27.78 | 0.7444 | 26.96 | 0.8109 | 31.27 | 0.9184 |
| ELAN Zhang et al. (2022b) | ×4 | 32.75 | 0.9022 | 28.96 | 0.7914 | 27.83 | 0.7459 | 27.13 | 0.8167 | 31.68 | 0.9226 |
| DLGSANet Li et al. (2023b) | ×4 | 32.80 | 0.9021 | 28.95 | 0.7907 | 27.85 | 0.7464 | 27.17 | 0.8175 | 31.68 | 0.9219 |
| IPT Chen et al. (2021) | ×4 | 32.64 | - | 29.01 | - | 27.82 | - | 27.26 | - | - | - |
| SwinIR Liang et al. (2021) | ×4 | 32.92 | 0.9044 | 29.09 | 0.7950 | 27.92 | 0.7489 | 27.45 | 0.8254 | 32.03 | 0.9260 |
| EDT Li et al. (2021a) | ×4 | 32.82 | 0.9031 | 29.09 | 0.7939 | 27.91 | 0.7483 | 27.46 | 0.8246 | 32.05 | 0.9254 |
| GRL-B* Li et al. (2023c) | ×4 | 32.90 | 0.9039 | 29.14 | 0.7956 | 27.96 | 0.7497 | 27.53 | 0.8276 | 32.19 | 0.9266 |
| SRFormer Zhou et al. (2023) | ×4 | 32.93 | 0.9041 | 29.08 | 0.7953 | 27.94 | 0.7502 | 27.68 | 0.8311 | 32.21 | 0.9271 |
| MambaIR Guo et al. (2024) | ×4 | 33.03 | 0.9046 | 29.20 | 0.7961 | 27.98 | 0.7503 | 27.68 | 0.8287 | 32.32 | 0.9272 |
| **Hi-Mamba-L** | ×4 | 33.05 | 0.9049 | 29.23 | 0.7966 | 28.01 | 0.7531 | 27.72 | 0.8296 | 32.43 | 0.9280 |
| **Hi-Mamba-L+** | ×4 | 33.08 | 0.9051 | 29.26 | 0.7969 | 28.02 | 0.7534 | 27.81 | 0.8304 | 32.56 | 0.9300 |

lightweight SR methods. Benefiting from the multi-scale mechanism and DA-HMG, Hi-Mamba-T and Hi-Mamba-S outperform SRFormer and MambaIR in terms of PSNR and SSIM across multiple benchmark datasets with comparable parameters and FLOPs. For example, compared to MambaIR, Hi-Mamba-S and Hi-Mamba-T reduce FLOPs by 294G and 390G, while improving the PSNR for ×2 SR on Urban100 by 0.21 dB and 0.12 dB, respectively. Meanwhile, Hi-Mamba-T significantly outperforms SRFormer by 0.24 dB on ×3 scale SR on Urban100, while reducing 25 GFLOPs and maintaining relatively consistent parameters.

**Qualitative Comparison.** In Fig. 3, we present the visual comparisons for ×4 SR. We can observe that previous CNN-based or Transformer-based methods suffer from blurry artifacts, distortions, and inaccurate texture restoration. In contrast, our method effectively reduces these artifacts, preserving more structural and clear details. More visual examples can be referred to in the Appendix. Moreover, as shown in Fig.4, we also visualize the Local Attribution Map (LAM) Gu & Dong (2021) to demonstrate the strong ability for long-range modeling using our Hi-Mamba-S.

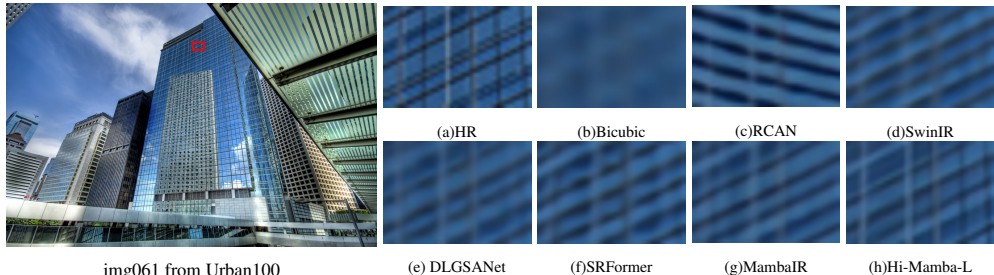

Figure 6: Qualitative comparison on the "img061" image of Urban100 for ×4 SR.

Table 4: Model complexity comparisons (×2). PSNR (dB) on Urban100 and Manga109, FLOPs, and Params are reported. Methods with "*" are replicated with standard settings.

| Method | EDSR | RCAN | SAN | HAN | NLSA | SwinIR | GRL-B* | MambaIR | Hi-Mamba-L |
|---|---|---|---|---|---|---|---|---|---|
| Params(M) | 43.09 | 15.59 | 15.87 | 63.61 | 41.80 | 11.90 | 20.20 | 20.42 | 21.58 |
| FLOPs(G) | 11,130 | 3,530 | 3,050 | 14,551 | 9,632 | 3,213 | 12,036 | 6,215 | 4,334 |
| PSNR-Urban100(dB) | 32.93 | 33.34 | 33.10 | 33.35 | 33.42 | 33.81 | 33.97 | 34.15 | 34.22 |
| PSNR-Managa109(dB) | 39.10 | 39.44 | 39.32 | 39.46 | 39.59 | 39.92 | 40.06 | 40.28 | 40.38 |

### 4.3 COMPARISON WITH PSNR-ORIENTED SR MODELS

To validate the scalability of Hi-Mamba, we further compare our Hi-Mamba-L with state-of-the-art PSNR-oriented SR models.

**Quantitative evaluations.** Tab. 3 summarizes the SR results at the scales of ×2 and ×4.

Our Hi-Mamba-L demonstrates superior performance compared to previous methods. In addition, the performance of Hi-Mamba-L can be further improved by using the self-ensemble strategy, denoted by Hi-Mamba-L+. For example, compared to SRFormer Zhou et al. (2023) and MambaIR Guo et al. (2024), our Hi-Mamba-L+ achieves significant PSNR gains of 0.42 dB and 0.21 dB on Manga109 for ×2 SR, respectively. For ×4 SR, our Hi-Mamba-L+ outperforms SRFormer by the PSNR of 0.18 dB and 0.35 dB on Set14 and Manga109, respectively. ×3 SR result is presented in the Appendix.

**Qualitative comparison.** We present the visual comparison of classic SR (×4) in Fig. 6. Compared with CNNs (e.g., RCAN) and Transformers (e.g., SwinIR, SRFormer, DLGSANet), as well as SSM-based MambaIR, Hi-Mamba reconstructs the most photo-realistic building texture compared to these models.

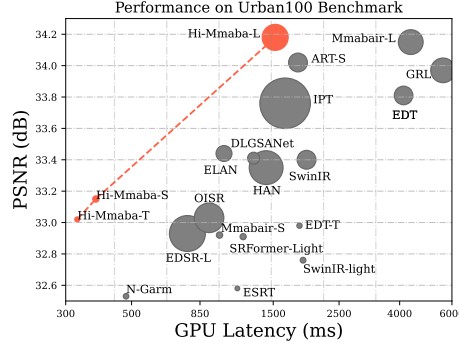

Figure 5: Performance on Urban100 for ×2 SR. The larger circles present larger computation costs on Params.

**Model Complexity Comparison.** Tab. 4 further makes our Hi-Mamba-L with CNNs and Transformers in terms of parameters and FLOPs. Our Hi-Mamba-L significantly reduces GFLOPs by 7,702 and 1,881, and achieves 0.22dB and 0.10dB PSNR gains on Manga109 over GRL-B and MambaIR. To evaluate the practical inference time, we conduct the experiments on the PSNR and speed results of different methods as shown in Fig. 5. We can observe that Hi-Mamba achieves the best latency-PSNR trade-off.

### 4.4 ABLATION STUDIES

In the ablation study, we train the models on DIV2K evaluated on Urban100 for 2× SR, as it contains images with rich structural details. For a fair comparison, we train the *baseline* composed of only L-SSM and MLP stacks with a depth number equal to Hi-Mamba-S.

**Ablation for key components of Hi-Mamba.** We first conduct the ablation study on the effect of R-SSM, G-FFN, and DA-HMG. As shown in Tab. 5, the R-SSM significantly improves PSNR by 0.19 dB. With the FFN replaced by G-FFN, this model achieves a gain of 0.04 dB over baseline+R-SMM while reducing 0.1M parameters and 15G FLOPs. Finally, by utilizing DA-HMG, we further improve

PSNR by 0.14 dB without incurring additional computational costs. This indicates that all the key components of Hi-Mamba show their effectiveness.

Table 5: Ablation study of the key components.

| R-SSM | G-FFN | DA-HMG | Params(M) | FLOPs(G) | PSNR | SSIM |
|-------|-------|--------|-----------|----------|------|------|
|       |       |        | 1.29      | 252      | 32.76 | 0.9339 |
| ✓     |       |        | 1.44      | 289      | 32.95 | 0.9354 |
| ✓     | ✓     |        | 1.34      | 274      | 32.99 | 0.9356 |
| ✓     | ✓     | ✓      | 1.34      | 274      | 33.13 | 0.9368 |

Table 7: Effect of fusion module in R-SSM.

| Fusion method | Upsampling | Repeat | Repeat $S_f = 0.5$ |
|---------------|------------|--------|--------------------|
| Params(M)     | 1.34       | 1.34   | 1.33               |
| FLOPs(G)      | 275        | 274    | 274                |
| GPU(ms)       | 387        | 379    | 371                |
| PSNR(dB)      | 33.06      | 33.13  | 33.08              |
| SSIM          | 0.9352     | 0.9368 | 0.9360             |

Table 6: Ablation study of DA-HMG.

| Model | PSNR(dB) | SSIM | Params(M) | FLOPs(G) |
|-------|----------|------|-----------|----------|
| Single-direction w/o alternation | 32.99 | 0.9356 | 1.34 | 274 |
| Two-direction alternation | 33.07 | 0.9361 | 1.34 | 274 |
| Four-direction alternation | 33.13 | 0.9368 | 1.34 | 274 |

Table 8: Abaliton of R-SSM channel number.

| #Channel | FLOPs(G) | Params(M) | PSNR(dB) |
|----------|----------|-----------|----------|
| 15       | 252      | 1.30      | 33.01    |
| 30       | 274      | 1.34      | 33.13    |
| 60       | 296      | 1.76      | 33.14    |

Table 9: Effect of region size in R-SSM.

| Region size | Params(M) | FLOPS(G) | GPU(ms) | PSNR(dB) | SSIM |
|-------------|-----------|----------|---------|----------|------|
| $1 \times 1$ | 1.52 | 312 | 662 | 33.13 | 0.9369 |
| $4 \times 4$ | 1.34 | 274 | 379 | 33.13 | 0.9368 |
| $8 \times 8$ | 1.34 | 271 | 365 | 33.01 | 0.9358 |

**Ablation for different scan modes in DA-HMG.** To investigate the effect of different scan modes in DA-HMG, we compare four-direction alternative scanning with single-direction without alternation(i.e., base-single), and two-direction alternative scanning, as summarized in Tab. 6. By default, single-direction without alternation only uses HMB-H, and two-direction alternative scanning uses HMB-H and HMB-V. We observe that the model using four-direction alternation achieves the best performance with an improvement of 0.14 dB PSNR and 0.06 dB PSNR over single-direction without alternation and two-direction alternation, respectively. Note that alternative direction scanning does not incur additional computational and memory costs. This indicates that direction alternation in DA-HMG can aggregate spatial information from different positions to improve reconstruction performance.

**Effect of fusion module in R-SSM.** As shown in Tab. 7, the repeat method implicitly incorporates the 2D spatial position information of features, achieving a PSNR of 0.07 dB higher than the upsampling method. It demonstrates the effectiveness of our 2D repeat fusion method. We also conduct additional ablation experiments on the learnable parameters $S_f$. We observed that the learnable parameter $S_f$ achieves only a slight increase of 0.01M parameters and 8ms GPU while outperforming fixed $S_f$ by 0.05dB PSNR. Thus, we default use the learnable $S_f$ for the fusion module.

**Ablation of R-SSM channel number.** We analyze the computational complexity of hierarchical design to achieve the best PSNR-FLOPs trade-off by changing the channel number of R-SSM. In Tab. 8, the channel number of 30 in R-SSM (*i.e.*, a half of L-SSM) achieves the best trade-off between performance and computation complexity.

**Ablation for the region size $n \times n$ in R-SSM.** As presented in Tab. 9, we find that a region patch size of 1×1 achieves the highest SSIM of 0.9369, but the inference time significantly increases, compared to patch sizes of 4×4 and 8×8. The region size of $4 \times 4$ yields the best trade-off between PSNR and inference speed. Thus, we set the region size to $4 \times 4$ for our experiments.

## 5 CONCLUSION

We present the Hierarchical Mamba Network (Hi-Mamba) in this paper for image super-resolution. Hi-Mamba is built on multiple-direction alternation hierarchical Mamba groups (DA-HMG), which allocates the isomeric single-direction scanning into cascading HMBs, enriching the modeling of spatial relationships. Each HMB consists of a Local SSM and a Region SSM, utilizing unidirectional scanning to aggregate multi-scale representations and enhance 2D spatial perception. Extensive experiments demonstrate that our Hi-Mamba has high potential compared to CNN-based and transformer-based methods.

## REPRODUCIBILITY STATEMENT

In this section, we provide a reproducibility statement for our proposed method. We detail the model architecture and core designs in Sec. 3, including the hierarchical mamba block (HMB) and Direction Alternation HMB Group (DA-HMG). Additionally, we present implementation details and elaborate on the experimental setup in Sec. 4.1. To ensure reproducibility, we will release the source code and pre-trained models. For more details, please refer to the Appendix.

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

# A APPENDIX

In this supplementary material, we provide additional information on the model analysis and experimental results of the Hierarchical Mamba (Hi-Mamba). In Section A.1, we report more information on the model details of different versions of Hi-Mamba. In Section A.2, We show the details of different scanning methods. In Section A.3, we present more extensive quantitative results on Hi-Mamba-L. In Section A.4, we provide additional qualitative comparisons. In Section A.5, we report the results of Hi-Mamba on other low-level vision tasks.

## A.1 MODEL DETAILS

We introduce Hi-Mamba model variants, scaled appropriately for different model sizes. We propose three sizes: Hi-Mamba-T (0.87M parameters), Hi-Mamba-S (1.34M parameters), and Hi-Mamba-L (22.58M parameters). Each Hi-Mamba variant is defined by its depth and width. Specifically, for Hi-Mamba-T and Hi-Mamba-S, there are 5 DA-HMGs, each containing 4 HMBs. The channel dimensions are set to 60 and 48 for Hi-Mamba-T and Hi-Mamba-S, respectively. The channel size for R-SSM is set to half of the respective channel dimensions. The channel expansion factor in G-FFN is set to 1.5. For Hi-Mamba-L, we increase the number of DA-HMGs to 7, with HMB numbers configured as [4, 4, 8, 8, 8, 4, 4]. Additionally, the channel dimension is increased to 180, and the expansion factor is set to 2. In addition, Hi-Mamba-L also introduces the channel attention mechanism. Other settings remain the same as Hi-Mamba-S. The model variants are summarized in the table 10.

Table 10: Hi-Mamba family. Channel Dimension indicates the number of channels for L-SSM/R-SSM.

| Method | Params(M) | FLOPs(G) | GPU(ms) | DA-HMG | HMB | Channel Dimension | Expansion Factor | Channel Attention |
|---|---|---|---|---|---|---|---|---|
| Hi-Mamba-T | 0.87 | 178 | 324 | 5 | [4, 4, 4, 4, 4] | 48/24 | 1.5 | No |
| Hi-Mamba-S | 1.34 | 274 | 379 | 5 | [4, 4, 4, 4, 4] | 60/30 | 1.5 | No |
| Hi-Mamba-L | 21.58 | 4334 | 1593 | 7 | [4, 4, 8, 8, 8, 4, 4] | 180/90 | 2 | Yes |

## A.2 COMPARISON OF DIFFERENT SCANNING METHODS IN SSM

In NLP tasks, data often contains causal relationships, so a base-single scanning approach is typically used, as shown in Fig. 7 (a). This single-directional scanning method has low complexity, but it is designed for text data and struggles to construct 2D spatial relationships in image tasks. VMamba Liu et al. (2024) and MambaIR Guo et al. (2024) designed the 2D-SSM scanning method for image tasks to model pixel information from various directions, as shown in Fig. 7 (b). 2D-SSM performs four different directional scanning sequences in parallel at each layer, achieving multi-directional parallelism. However, in visual SSM architectures, parallel scanning across four sequences significantly increases computational overhead, making it less efficient for processing high-resolution images. This compromises the speed advantage of the Mamba architecture. To address this issue, we designed the DA-HMG, as shown in Fig. 7 (c). In each layer, only a single directional scanning sequence is computed. Compared to the base-single approach, the use of different directional scanning sequences in DA-HMG allows pixels to integrate neighboring information from multiple directions. In contrast to 2D-SSM, DA-HMG selects only one scanning direction per layer, significantly improving inference speed while reducing computational and parameter overhead.

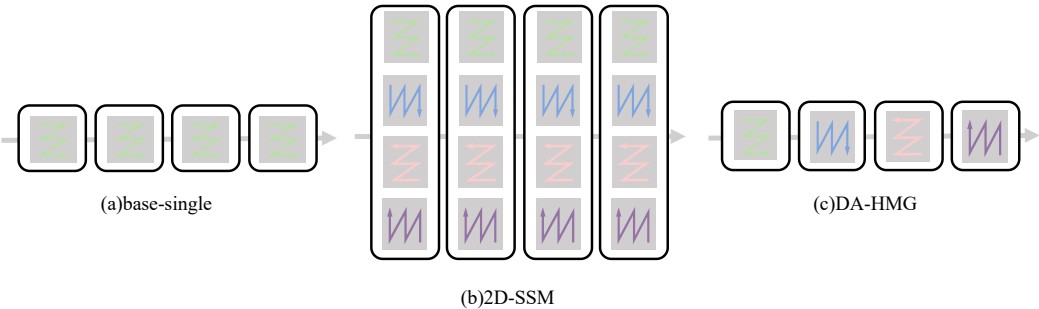

Figure 7: Visualization of different scanning methods.

### A.3 MORE EXTENSIVE QUANTITATIVE RESULTS

We further report the results compared to PSNR-oriented SR Models at scale ×3. In the main paper, we have evaluated the effectiveness for ×2 and ×4 SR performance. As shown in Tab. 11, Hi-Mamba-L achieves the best PSNR across all datasets, surpassing the second-place MambaIR by 0.11 dB and the transformer-based method SRformer by 0.28 dB on Manga109 with comparable parameter and FLOPs.

Table 11: Comparison of different PSNR-oriented SR models on five benchmarks with scale ×3.

| Model | Scale | Set5 | | Set14 | | BSD100 | | Urban100 | | Manga109 | |
|---|---|---|---|---|---|---|---|---|---|---|---|
| | | PSNR | SSIM | PSNR | SSIM | PSNR | SSIM | PSNR | SSIM | PSNR | SSIM |
| EDSR Lim et al. (2017) | ×3 | 34.65 | 0.9280 | 30.52 | 0.8462 | 29.25 | 0.8093 | 28.80 | 0.8653 | 34.17 | 0.9476 |
| RCAN Zhang et al. (2018) | ×3 | 34.74 | 0.9299 | 30.65 | 0.8482 | 29.32 | 0.8111 | 29.09 | 0.8702 | 34.44 | 0.9499 |
| SAN Dai et al. (2019) | ×3 | 34.75 | 0.9300 | 30.59 | 0.8476 | 29.33 | 0.8112 | 28.93 | 0.8671 | 34.30 | 0.9494 |
| HAN Niu et al. (2020) | ×3 | 34.75 | 0.9299 | 30.67 | 0.8483 | 29.32 | 0.8110 | 29.10 | 0.8705 | 34.48 | 0.9500 |
| IGNN Zhou et al. (2020) | ×3 | 34.72 | 0.9298 | 30.66 | 0.8484 | 29.31 | 0.8105 | 29.03 | 0.8696 | 34.39 | 0.9496 |
| CSNLN Mei et al. (2020) | ×3 | 34.74 | 0.9300 | 30.66 | 0.8482 | 29.33 | 0.8105 | 29.13 | 0.8712 | 34.45 | 0.9502 |
| NLSN Mei et al. (2021) | ×3 | 34.85 | 0.9306 | 30.70 | 0.8485 | 29.34 | 0.8117 | 29.25 | 0.8726 | 34.57 | 0.9508 |
| ELAN Zhang et al. (2022b) | ×3 | 34.90 | 0.9313 | 30.80 | 0.8504 | 29.38 | 0.8124 | 29.32 | 0.8745 | 34.73 | 0.9517 |
| DLGSANet Li et al. (2023b) | ×3 | 34.95 | 0.9310 | 30.77 | 0.8501 | 29.38 | 0.8121 | 29.43 | 0.8761 | 34.76 | 0.9517 |
| IPT Chen et al. (2021) | ×3 | 34.81 | - | 30.85 | - | 29.38 | - | 29.49 | - | - | - |
| SwinIR Liang et al. (2021) | ×3 | 34.97 | 0.9318 | 30.93 | 0.8534 | 29.16 | 0.8145 | 29.75 | 0.8826 | 35.12 | 0.9537 |
| EDT Li et al. (2021a) | ×3 | 34.97 | 0.9316 | 30.89 | 0.8527 | 29.44 | 0.8142 | 29.72 | 0.8814 | 35.13 | 0.9534 |
| GRL-B* Li et al. (2023c) | ×3 | 35.05 | 0.9323 | 31.00 | 0.8543 | 29.49 | 0.8153 | 29.83 | 0.8837 | 35.24 | 0.9541 |
| SRFormer Zhou et al. (2023) | ×3 | 35.02 | 0.9323 | 30.94 | 0.8540 | 29.48 | 0.8156 | 30.04 | 0.8865 | 35.26 | 0.9543 |
| MambaIR Guo et al. (2024) | ×3 | 35.08 | 0.9323 | 30.99 | 0.8536 | 29.51 | 0.8157 | 29.93 | 0.8841 | 35.43 | 0.9546 |
| **Hi-Mamba-L** | ×3 | 35.10 | 0.9329 | 31.08 | 0.8545 | 29.54 | 0.8173 | 29.99 | 0.8846 | 35.47 | 0.9548 |
| **Hi-Mamba-L+** | ×3 | 35.15 | 0.9332 | 31.16 | 0.8547 | 29.55 | 0.8176 | 30.08 | 0.8871 | 35.54 | 0.9553 |

### A.4 MORE VISUALIZATION RESULTS

We add more visualization results of our Hi-Mamba-S and Hi-Mamba-L, making comparisons with other models on different datasets for ×4 SR in Fig. 8, and Fig. 9, respectively. In img078, ESRT and MambaIR exhibit artifacts in their restorations, whereas Hi-Mamba-S successfully recovers the structural detail contours.

### A.5 MORE IMAGE RESTORATION TASKS

We further evaluate the performance of our Hi-Mamba on image deblurring with state-of-the-art image deblurring models. including SRN Tao et al. (2018), DMPHN Zhang et al. (2019a), MPRNet Zamir et al. (2021), IPT Chen et al. (2021), Restormer Zamir et al. (2022), Uformer Wang et al. (2022) and Stripformer Tsai et al. (2022) In Tab. 12, our Hi-Mamba achieves the highest 33.27dB PSNR, outperforming the previous best SOTA Stripformer with 0.19dB.

Table 12: Quantitative evaluations on GoPro dataset for image deblurring.

| Method | SRN | DMPHN | MPRNet | IPT | Restormer | Uformer | Stripformer | Hi-Mamba |
|---|---|---|---|---|---|---|---|---|
| PSNR | 30.26 | 31.20 | 32.66 | 32.52 | 32.92 | 33.06 | 33.08 | 33.27 |
| SSIM | 0.9342 | 0.9453 | 0.9589 | - | 0.9611 | 0.9670 | 0.9624 | 0.9651 |
| Params(M) | 6.8 | 21.7 | 20.1 | 114 | 26.1 | 50.9 | 19.7 | 23.6 |

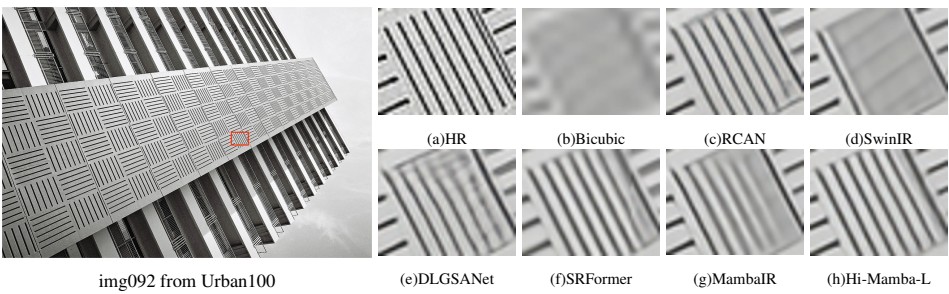

| | |
|---|---|
| img092 from Urban100 | (a)HR (b)Bicubic (c)RCAN (d)SwinIR (e)DLGSANet (f)SRFormer (g)MambaIR (h)Hi-Mamba-L |

Figure 8: Qualitative comparison on the "img092" image of Urban100 for ×4 classical SR.

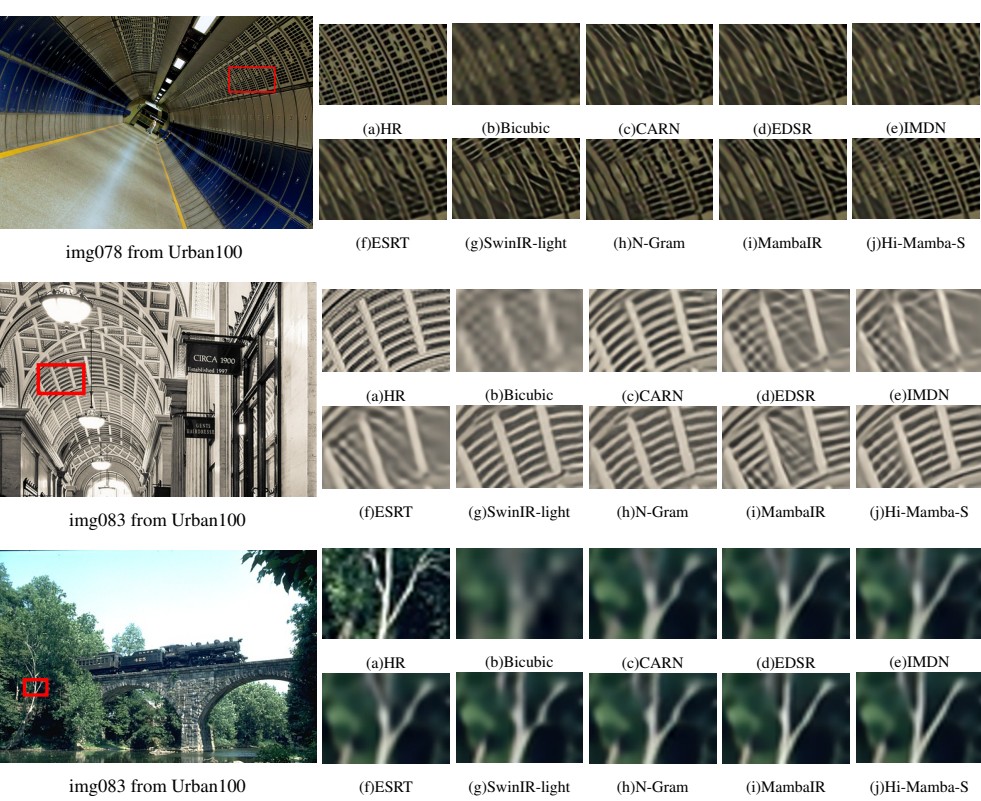

Figure 9: More qualitative comparisons for ×4 SR.

