# OpenReview forum: "Hi-Mamba: Hierarchical Mamba for Efficient Image Super-Resolution"
_ICLR.cc/2025/Conference — ICLR 2025 Conference Withdrawn Submission_

### Official Review · Reviewer_fUus · 2024-11-01

**Soundness:** 2
**Presentation:** 3
**Contribution:** 2
**Rating:** 5
**Confidence:** 4

**Summary:**

The paper proposes a novel hierarchical architecture, Hi-Mamba, designed to improve image super-resolution using a structured state space model approach. The architecture introduces the Hierarchical Mamba Block, consisting of Local SSM  and Region SSM components to achieve multi-scale visual context modeling with single-direction scanning. Additionally, a Direction Alternation Hierarchical Mamba Group is used to model spatial relationships by alternating scan directions. The proposed method aims to address the high computation cost associated with multi-direction scanning in existing Mamba-based models. Results on several SR benchmarks show improvements in PSNR and computational efficiency.

**Strengths:**

1. The Hi-Mamba architecture introduces a hierarchical structure and a new directional alternation technique, which is a creative approach to enhancing Mamba's performance in SR tasks.

2. The paper addresses high computation costs problem of SSM-based SR models due to multi-directional scanning, and provides an alternative that maintains performance with less computational demand.

**Weaknesses:**

1. While the proposed changes are effective, they appear as relatively incremental modifications over existing SSM-based models, particularly MambaIR. The hierarchical structure and directional alternation are valuable but do not constitute a fundamentally new approach, which may limit the overall impact.

2. The performance improvement of Hi-Mamba compared to other state-of-the-art models, such as MambaIR and Transformer-based SR models, is relatively small. While Hi-Mamba demonstrates slight gains in PSNR and SSIM across various benchmarks, the marginal advantages may not be compelling enough to justify its adoption over other established models, particularly in applications where computational efficiency is a critical factor.

3. The paper lacks a thorough analysis of the complexity trade-offs introduced by the hierarchical structure and directional alternation. The reported reduction in computational complexity is promising, but additional comparisons or ablations on how the design changes impact efficiency, memory usage, and latency would strengthen the paper.

4. While Hi-Mamba shows slight improvements over MambaIR and Transformer-based methods, the paper could be more comprehensive in its comparison against a broader range of state-of-the-art SR methods. Including additional evaluation metrics or visual comparisons on challenging datasets would provide a clearer picture of Hi-Mamba's effectiveness in real-world scenarios.

**Questions:**

1. How does the directional alternation in DA-HMG specifically enhance spatial relationship modeling, and have other combinations of scan directions been tested to assess their impact on performance? A more detailed explanation or visualization of how these directional alternations contribute to improved spatial modeling would be helpful.

2. Could the authors provide a more detailed comparison of Hi-Mamba’s computational trade-offs, particularly in memory usage and inference latency, relative to other state-of-the-art methods like MambaIR and Transformer-based models?

---

### Official Review · Reviewer_kVNh · 2024-11-02

**Soundness:** 3
**Presentation:** 3
**Contribution:** 2
**Rating:** 3
**Confidence:** 4

**Summary:**

State Space Models (SSM), such as Mamba, demonstrate powerful representation capabilities in modeling long-range dependencies. However, their sequential nature necessitates multiple scans in different directions to compensate for the loss of spatial dependencies when flattening images into 1D sequences. This multi-directional scanning strategy significantly increases computational overhead. This paper proposes a new hierarchical Mamba network, Hi-Mamba, for image super-resolution. Hi-Mamba incorporates two key designs: (1) a Hierarchical Mamba Block (HMB) composed of Local SSMs (L-SSM) and Regional SSMs (R-SSM) with single-direction scanning, which aggregates multi-scale representations to enhance contextual modeling capabilities; (2) Direction-Alternating Hierarchical Mamba Groups (DA-HMG) that assign equivalent single-direction scans to cascaded HMBs, enriching spatial relationship modeling. The model is trained on two commonly used datasets: DIV2K and Flicker2K. It is evaluated on five standard super-resolution (SR) benchmark datasets: Set5, Set14, BSD100, Urban100, and Manga109, tested at scaling factors of ×2, ×3, and ×4. Ablation experiments are conducted for different scanning modes, fusion module effects, R-SSM channel numbers, and region sizes of n×n. Extensive experiments demonstrate that Hi-Mamba exhibits superior performance across the five benchmark datasets for efficient SR.

**Strengths:**

Dual feature extraction capability: By combining the L-SSM and R-SSM modules, Hi-Mamba can extract local and regional features of the image at the same time, focusing on small-scale and large-scale contextual relationships respectively, thereby enhancing the expression ability of features.

Superior performance and computational efficiency: Experimental results show that Hi-Mamba performs well on multiple benchmark data sets. For example, in the ×2 SR task of the Urban100 data set, Hi-Mamba-S can significantly reduce the amount of calculation (294G FLOPs) while improving PSNR.

Higher visual quality: In the ×4 SR task, Hi-Mamba can reduce blur artifacts and distortion more effectively than traditional CNN or Transformer models, retain more clear details and structural information, and improve the visual quality of the image. quality.

**Weaknesses:**

1. The proposed Hi-Mamba-T and Hi-Mamba-S variants do not show significant differences in PSNR and SSIM evaluation metrics compared to Hi-Mamba-T, aside from the differences in parameters and FLOPs. The advantages and roles of the two variants are not clearly stated.
2. As the model variants Hi-Mamba-T, Hi-Mamba-S, and Hi-Mamba-L increase in depth and width, their parameters and FLOPs also increase. When using more complex models, the optimization performance in terms of computation may not be evident.
3.Some newer super-resolution methods such as DAT and HAT are not compared in the article. Compared with these methods, the method proposed in the article does not have a great advantage.
4.Although the improvement of the mamba variant structure used by the author has achieved certain results, it still looks relatively simple.

**Questions:**

My questions can refer to the content mentioned in the shortcomings, and the author can answer them one by one according to the shortcomings mentioned.

---

### Official Review · Reviewer_6h8G · 2024-11-03

**Soundness:** 3
**Presentation:** 3
**Contribution:** 1
**Rating:** 3
**Confidence:** 5

**Summary:**

This work proposes a hierarchical Mamba network for efficient image super-resolution. The fusion of Local SSM and Region SSM enables improved contextual modeling. Additionally, the authors proposed a DA-HMG block to further enrich the spatial relationship modeling capabilities. The extensive experiments validate the effectiveness of the proposed model.

**Strengths:**

1. The paper is well-structured and easy to follow. The figures provide clear structure information about different key components of the proposed method. Moreover, the figures are clean and well-organized, with appropriate color schemes that enhance clarity and readability.

2. The visual comparisons on the Urban100 benchmark demonstrate the effectiveness of the proposed method. Hi-Mamba restores better local textures than competitive methods, while maintaining higher LAM attributions and area of contributions.

**Weaknesses:**

1. The motivation for this paper largely draws from previous works, such as MambaIR [1], LocalMamba [2], Vmamba [3], and MSVMamba [4], with limited innovation. First, the introduction section does not adequately elaborate on the paper’s motivation. For instance, the reasons why Local SSM and Region SSM enhance modeling capabilities are not sufficiently explained. Additionally, using Local SSM and multi-scale features is not novel within Mamba-related studies (e.g., LocalMamba and MSVMamba).

2. The directional alternation mentioned in the paper is also not a new concept; similar alternating scanning is discussed in MambaCSR [5]. While this scanning approach can reduce computation to some extent, it also introduces certain performance trade-offs which can be further discussed.

[1] VMamba: Visual State Space Model, Yue Liu et al. CoRR abs/2401.10166.

[2] Guo H, Li J, Dai T, et al. Mambair: A simple baseline for image restoration with state-space model[C]//European Conference on Computer Vision. Springer, Cham, 2025: 222-241.

[3] Huang T, Pei X, You S, et al. Localmamba: Visual state space model with windowed selective scan[J]. arXiv preprint arXiv:2403.09338, 2024.

[4] Shi Y, Dong M, Xu C. Multi-Scale VMamba: Hierarchy in Hierarchy Visual State Space Model[J]. arXiv preprint arXiv:2405.14174, 2024.

[5] Ren Y, Li X, Guo M, et al. MambaCSR: Dual-Interleaved Scanning for Compressed Image Super-Resolution With SSMs[J]. arXiv preprint arXiv:2408.11758, 2024.

**Questions:**

1. The paper introduces region SSM and local SSM. However, from a theoretical perspective, how do these designs enhance contextual information modeling to achieve performance gains? It would be better to first address this problem in section 3.3.1, rather than directly introducing the structure of the block.
2. Are there any visual comparison results other than the Urban100 benchmark?
3. Minor: it seems that Figure 6(c) is mistakenly placed.

---

### Official Review · Reviewer_Nx39 · 2024-11-03

**Soundness:** 2
**Presentation:** 2
**Contribution:** 2
**Rating:** 3
**Confidence:** 5

**Summary:**

This work focuses on improving the performance of Mamba architecture in image super-resolution task. It proposes to enhance long-range modeling capability using Hierarchical Mabma Block. Meanwhile, mamba blocks with various scanning direction are employed to further enrich the spatial information. Three models with different size are introduced, with extensive experimental results to verify its superiority compared to previous Mamba- and Transformer-based methods under comparable or less computational costs.

**Strengths:**

1. Various techniques are proposed to improve the modeling ability of Mamba-based SR model, with comprehensive ablation studies validating the effectiveness.
2. The Hi-Mamba-L strikes a better balance between performance and computational cost compared to the previous mamba-based method MambaIR.

**Weaknesses:**

1. The results of SRFormer are presented in Table 3, however, its model complexity and GPU Latency results are not included in Table 4 and Figure 5.
2. Most citations that be in parenthesis are not in parenthesis, which impacts the readibility.
3. The novelty seems limited.

**Questions:**

1. Have you experimented replacing the Avgpool in Region SSM with other downsampling methods, e.g., strided conv.
2. In what aspects do you think the mamba-based methods perform better compared to the Transformer-based methods. As far as I know, several recent Transformer-based HAT, ATD, IPG have achieved much outstanding results, could you please make a comparison with them.
3. Have you compared the LAM visualization of Hi-Mamba with GRL-B and SRFormer, since they have an enlarged window-size compared to SwinIR.

---

### Note · Authors · 2024-11-25

I have read and agree with the venue's withdrawal policy on behalf of myself and my co-authors.